# *In Vitro* Hepatic Models to Assess Herb–Drug Interactions: Approaches and Challenges

**DOI:** 10.3390/ph16030409

**Published:** 2023-03-08

**Authors:** Hlengwa N., Masilela C., Mtambo T. R., Sithole S., Naidoo S., Machaba K. E., Shabalala S. C., Ntamo Y., Dludla P. V., Milase R. N.

**Affiliations:** 1Department of Biochemistry and Microbiology, University of Zululand, KwaDlangezwa 3886, South Africa; 2Department of Biochemistry, North-West University, Mafikeng 2745, South Africa; 3Biomedical Research and Innovation Platform and South African Medical Research Council, Cape Town 7505, South Africa

**Keywords:** herb–drug interactions, hepatic models, human liver microsomes, S9 liver fractions, primary hepatocytes, precision liver slices, 3D culture, pharmacokinetics

## Abstract

A newfound appreciation for the benefits of herbal treatments has emerged in recent decades. However, herbal medication production still needs to establish standardized protocols that adhere to strict guidelines for quality assurance and risk minimization. Although the therapeutic effects of herbal medicines are extensive, the risk of herb–drug interactions remains a serious concern, limiting their use. Therefore, a robust, well-established liver model that can fully represent the liver tissue is required to study potential herb–drug interactions to ensure herbal medicines’ safe and effective use. In light of this, this mini review investigates the existing *in vitro* liver models applicable to detecting herbal medicines’ toxicity and other pharmacological targets. This article analyzes the benefits and drawbacks of existing *in vitro* liver cell models. To maintain relevance and effectively express the offered research, a systematic strategy was employed to search for and include all discussed studies. In brief, from 1985 to December 2022, the phrases “liver models”, “herb–drug interaction”, “herbal medicine”, “cytochrome P450”, “drug transporters pharmacokinetics”, and “pharmacodynamics” were combined to search the electronic databases PubMed, ScienceDirect, and the Cochrane Library.

## 1. Introduction

It is already acknowledged that almost 80% of the world’s population depends on traditional herbal medicine for primary health care [1]. Even worse, the COVID-19 pandemic has exacerbated the use of herbal drugs in developed and developing countries [2,3]. Conversely, this has been accompanied by a staggering increase in herb–drug interactions (HDI), aggravating the demand to find robust models to assess and predict HDI. Herb–drug interactions are an essential aspect of clinical therapeutic effectiveness. According to a study that appeared in the *Journal of the American Medical Association*, adverse medication events are responsible for an estimated 100,000 deaths in the United States alone each year [4]. While this encompasses all forms of adverse medication events, herb–drug interactions may play a role. Another study published in the *Journal of Clinical Pharmacology* revealed that in the United States, herb–drug interactions are responsible for around 30% of adverse drug reactions [5].

Herb–drug interactions alter the pharmacokinetics or pharmacodynamics of pharmaceutical drugs, resulting in therapeutic failure or toxicity. The key mechanisms responsible for the majority of pharmacokinetic interactions are the modulation of the activity of metabolic enzymes, especially cytochrome P450 enzymes (CYPs). Comparatively less research has been conducted on herb–drug interactions in pharmacodynamics than in pharmacokinetics. This area should receive additional focus and effort. Herb–drug interactions are complex, and therapeutic consequences may result from both pharmacokinetic and pharmacodynamic interactions. Different factors, such as the combined use of drugs, the source of the herbal medicines, the patient’s unique characteristic features, and the constituents or dosages of various regimens, all contribute to the complications surrounding herb–drug interactions [6]. Lack of understanding regarding potential herb–drug interactions will pose a significant threat to the safety of individuals receiving medical care. Contrariwise, certain interactions may be medically advantageous. In the future, these interactions may be used to develop novel therapeutic strategies. As a result, this issue has highlighted the prerequisite for establishing pre-clinical models that could be used to understand the implications of herbal–drug interactions.

As such, the application of *in vitro* liver models for herb–drug interaction studies to validate the possibility of adverse and toxic effects that arise when conventional drugs are mixed with herbal medicine is a longstanding practice [7]. This is mainly due to the expression of drug-metabolizing cytochrome P450 enzymes, associated transporters, and efflux proteins within these liver models (Figure 1). The currently available liver models recommended by the Food and Drug Administration have inherent limitations; hence, they must be combined to elucidate the mechanisms of interactions fully. This is time-consuming and expensive, and the difficulties inherent in the phenotypic and genotypic characteristics of these *in vitro* liver models, including interlaboratory protocol variations, have resulted in irreproducible results. These limitations affect the extrapolation of data from pre-clinical research involving herb–drug interactions. Hence, different approaches to enhancing the physiological relevance of hepatic *in vitro* systems are being pursued to improve the prediction of herb–drug interactions.

For example, applying three-dimensional (3D) culture as a novel experimental methodology using human cell lines is emerging as a relevant approach to predicting potential herb–drug interactions [8]. Indeed, in experimental pharmacology, there is a gradual shift from animal experimentation to more predictive *in vitro* models, including 3D culture, and liver organ-on-chip models. Liver organ-on-chip models are miniature platforms that aim to mimic the physiological and biochemical functions of the liver *in vitro* [9]. These models provide a unique opportunity for studying liver functions, toxicity, herb–drug interactions, and disease mechanisms in a controlled environment [10].

These *in vitro* models can be produced from human cells, which express the necessary cytochrome P450 (CYP) metabolizing enzyme isoforms for pharmacokinetic research that is frequently lacking in animal models [11]. A classic example is the use of Wistar rats for pharmacokinetic studies, where the rodent model expresses CYP3A1 and CYP3A2 [12] instead of CYP3A4, which is the most abundant CYP in humans and responsible for metabolizing approximately 50–60% of xenobiotics [13]. Given the limitations of the current models used in herb–drug interactions studies, as well as the urgent need to provide clinically relevant knowledge for healthcare professionals, the importance of innovative models that express not only CYP enzymes, but also uptake and efflux transporters cannot be overstated [14].

As a result, this mini-review looks at the current *in vitro* liver models that can be used to determine the toxicology of herbal medicines and other drug targets. This review addresses the limitations and advantages/applications of the available *in vitro* liver cell models. This is still necessary for developing further, well-defined alternative models that can be used independently to unravel the mechanisms of herb–drug interactions. 

## 2. Research Methodology and Search Strategy

To ensure uniformity, all reviewers screened the same 31,150 publications, reviewed the results, and revised the screening and data extraction methodology before beginning screening for this study. Each article in our searches for possibly relevant publications had its title, abstract, and full text reviewed by a team of ten reviewers working in pairs. We resolved differences in research selection and data extraction through consensus and, if necessary, discussion with other reviewers. The review’s selection criteria were based on experimental data that demonstrated significant limitations for each *in vitro* model. Studies indicating modified, ultrastructural specialization, stability, and altered expression of critical xenobiotic-involved genes models were included. Studies involving recombinant cytochrome P450 enzymes were excluded since they were not obtained directly from the liver and fell outside this review’s scope. Studies on liver cytosolic fractions were also excluded due to the limited data currently available on the model’s utilization due to the lack of cytochrome P450 enzymes and other essential genes in xenobiotic metabolism. A total of 42 papers were included in this review. More details with regards to the selection of research articles are shown in Figure 2. 

A systematic approach was used to search for and include all discussed studies to remain relevant and clearly articulate the presented literature. In a nutshell, the terms “liver models”, “herb–drug interaction”, “herbal medicine”, “cytochrome P450”, “drug transporters pharmacokinetics”, and “pharmacodynamics” were combined to search the electronic databases of PubMed, ScienceDirect, and the Cochrane Library from 1980 to December 2022. The results are discussed in the subsequent sections on the relevance of various *in vitro* liver models, including human liver microsomes (HLM) and S9 fractions, precision-cut liver slices, and primary or immortalized (HepG2/C3A cells).

## 3. Overview of Liver Models Used in Herb–Drug Interaction Studies

Various systems have been established to simulate liver metabolism and understand herb–drug interactions, including human liver microsomes (S9 liver fractions), hepatocytes, and precision-cut liver slices, amongst others (Appendix A). Additionally, the expression of phase I and phase II enzyme profiles and the longevity of the aforementioned vary across each of these notable models.

## 4. Recombinant Cytochrome P450 Enzymes

Recombinant cytochrome P450 (P450) enzymes usually offer an alternative *in vitro* system for predicting human metabolic clearance. Consequently, recombinant enzymes are typically utilized as prediction models to evaluate induction experiments in order to evaluate potential herb–drug interactions. The known advantages of these recombinant enzymes include the ability to account for differences in P450 expression between people and the value of getting a head start on understanding the enzymology of drug metabolism. The clearance prediction accuracy of this *in vitro* system needs to be tested with a large set of reference drugs [15]. Systems for the heterologous expression of recombinant P450 enzymes include expression in bacterial cells, expression in yeast cells, mammalian expression systems, and baculovirus-driven expression in insect cells [13]. Usually, recombinant enzymes are used to supplement the shortfalls that normal cell-based systems have when it comes to the expression of cytochrome P450 enzymes to assess HDI [16,17,18]. This is most possibly due to the stability of recombinant cDNA-expressed cytochrome P450 enzymes. A great success in the use of baculosomes from insect cells (amongst many other expression systems including bacteria, yeast, and mammalian cells) for heterologous expression of recombinant P450 enzymes has been observed in the past. However, to make accurate predictions about metabolic clearance with cDNA-expressed P450 enzymes, it is best to compare the activity of P450 in recombinant systems with that in human liver microsomes (HLMs), S9 liver fractions, and cytosolic liver fractions.

## 5. Human Liver Microsomes (HLM), S9 Fractions, and Cytosolic Liver Fractions

Membrane fragments such as human liver microsomes, S9 liver fractions, and cytosolic liver fractions are extensively used in liver models in pharmacology [19]. Microsomes are derived from subcellular fractionations of tissue with endoplasmic reticulum enrichment obtained from procurement organizations donated by various donors [20]. This system provides the most convenient way to study CYP-mediated HDI because it is readily available at a low cost with abundant membrane-bound CYP enzymes. These models are extensively used to conduct clearance experiments to assess potential herb–drug interactions. Nonetheless, the limitation of using this model is the absence of some major phase II metabolizing enzymes, although the HLM is regarded as a high-throughput *in vitro* screening model. Other limitations are linked to manufacturers’ preparation protocol; for example, two suppliers of rat liver microsomes metabolized buspirone and loperamide well, whereas the third vendor showed no activity, and three batches from the same vendor showed varying activity. Animal-to-animal variance and vendor preparation techniques may explain microsomal activity discrepancies. Some vendors employed phenylmethylsulfonylfluoride to prepare liver microsomes because it inhibits trypsin-like proteases that could degrade microsomes. Others used EDTA. Phenylmethylsulfonyl fluoride inhibits carboxylesterases, and ethylenediaminetetraacetic acid (EDTA) chelates calcium and iron, inhibiting calcium-dependent phospholipases and lipid peroxidation. To compare and justify study results across batches, verify the vendor’s microsomal characterization data for cytochrome b5, P450, and NADPH-cytochrome c reductase activity. Therefore, their application cannot be relied on as a representative system on its own for *in vitro* drug metabolism investigations. Hence, they are regularly used as a predictive model [21]. Microsomes significantly lack cytosolic enzymes, despite containing the endoplasmic reticulum subcellular fraction, which predominantly includes cytochrome P450s and uridine 5’-diphospho-glucuronosyltransferases (UGTs) (Table 1) [21]. Therefore, S9 fractions are utilized to play a supplementary role when these two models are used in combination.

S9 fractions are post-mitochondrial supernatant fractions containing a mixture of microsomes and cytosol [20]. S9 fractions also contain a wide variety of phase I and phase II enzymes. This feature makes S9 fractions suitable for determining drugs’ pharmacokinetic profiles as well as HDI in humans [22]. Metabolic reactions that occur in the liver microsomes can be confirmed using S9 fractions. However, the limiting factors of this model are the intra-subject differences in drug-metabolizing enzyme activities and sensitivity (Table 1) [20], consistent with any cell-free system; the downsides include the probable inactivation or absence of certain enzymes, such as flavin-monooxygenases (FMOs), the loss of cellular compartmentalization, and the requirement to add cofactors during incubation. In addition to the apparent inheritance of dilution enzymes and low translatability, S9 fractions exhibit a significant level of cytotoxicity in cell-based experiments.

Another popular liver fractionation model is cytosolic liver fractions. They are derived by differential centrifugation of homogenates. The soluble phase I enzymes, such as esterases, amidases, or epoxyde hydrolases, are found in the liver cytosolic fraction. Soluble phase II enzymes, such as most of the sulfotransferases (ST), glutathione s-transferases (GST), and N-acetyltransferases (NAT), are found in the liver cytosolic fraction (NAT). Some exogenous cofactors, such as adenosine 3′-phosphate-5′-phosphosulfate (PAPS) for sulfotransferase activity 2, can be added to the catalytic activity of phase II enzymes, particularly in concentrated (ultrafiltration) cytosolic fractions. Although this cytosolic fraction cannot be used as a whole metabolic system, it can aid in the resolution of several challenges in metabolic profiling for medicines processed by soluble enzymes. It is also possible to obtain a more complete system by co-incubating with microsomes [23] or using them with alternative models such as primary hepatocytes to fully assess herb–drug interactions. Although these models are suitable candidates, native limitations warrant that another model is used to validate the findings.

**Table 1 pharmaceuticals-16-00409-t001:** The advantages and disadvantages of using each *in vitro* hepatic model for drug testing/hepatotoxicity and new/future developments.

Type of Liver Model	Advantages	Disadvantages/Limitations	Advancements/What Is New in the Field?	References
**PHH**	Maintain original structure and liver-specific functions *in vivo*, “gold standard.”	Significant batch-to-batch variation, limited availability, short lifespan.	Stem cell-derived hepatocytes.	[24,25]
**PCLS**	Contains all the cells of liver tissue in their natural environment.	Fierce competition for organ donors in research, short lifespan.	Long-term PCLS as “pre-clinical test.”	[26,27]
**S9 Fractions**	High throughput *in vitro* screening model, more readily available than hepatocytes.	Lack of major phase II metabolizing enzymes.	Organoids.	[21]
**HLM**	Model for high-throughput *in vitro* screening, greater availability than hepatocytes.	Lack of stability for long-term culture.	Organoids.	[21]
**LFC**	*In vitro* high-throughput screening model, greater availability than hepatocytes.	Lack of major phase I and phase II metabolizing enzymes.	Organoids.	[23,27]
**HEPG2/C3A cells**	Easily attainable, cost effective.	2D monocultures show low expression of major CYPs.	3D culture HEPG2/C3A liver spheroids.	[28]
**3D culture liver spheroids**	Reproducibility of results from long-term drug/herbal treatment.	3D spheroids expert skill.	3D bioprinting technology/ liver organ-on-chip.	[9,24,29]
**Liver organ-on-chip models**	Real-time monitoring, high-throughput screening, cost-effective when compared to animals.	Limited functionality, short lifespan, lack of standardization.	3D bioprinting technology.	[30,31,32]

Abbreviations: PHH = primary human hepatocytes, PCLS = precision-cut liver slices, HLM = human liver microsomes, LCF = liver cytosolic fraction.

## 6. Precision-Cut Liver Slices

Precision-cut liver slices (PCLS) are an ex vivo liver model that contains all the liver tissue cells in their natural environment, demonstrating intracellular and cell matrixes [26]. Due to the availability of metabolizing enzymes, this model, which is derived from a single donor, is frequently used to study liver metabolism, toxicity of xenobiotics, and HDI [31,33]. The most prevalent issue is the high competition for organ donors. Human donor tissue is scarce, and more importantly, the quality of the liver tissue is highly variable from one donor to the next, affecting the reproducibility of results (Table 1). For example, a study by Dewyse et al. demonstrated the variations in basal cytochrome enzyme levels in human liver slices from different human donors, which could vary up to 500-fold. These differences affect inter-individual responses to drug- or herb-induced toxicity, making it challenging to predict herb–drug interactions. Another significant drawback of using this hepatic cell model is the rapid liver damage and cell death that happens (short-term viability of only 4–6 days post-tissue coring due to inadequate oxygenation and nutrition), which ultimately has a cascading effect on inducing repair and regenerative responses, resulting in fibrosis once the slices are cultured [26]. Innovative experimental methods to circumvent this limitation include perfusion of human-derived PCLS before slicing further to maintain the viability and functionality of the slices, as previously discussed [34]. However, further studies are required to perfect this methodology. Other limitations of this model include low stability and production of phase II metabolizing enzymes, which are essential for herb–drug interaction studies [34]. Nonetheless, PCLS remains a well-established alternative model extensively used for hepatic metabolism without separating cells and keeping the natural cellular environment with a complete metabolic program [27]. It is deemed that if there can be a standard method established to prolong the lifespan of the slices efficiently, PCLS can be the next “gold standard”, with 4099 published articles. Other challenges include limited availability and inter-donor functional and genetic variability. Furthermore, the donated tissue is often extracted from the damaged liver, which has been removed due to a particular disease infestation, such as cancer, and may have other existing pathologies that could potentially induce tissue damage [28]. This restricts their application and suitability for herb–drug interaction studies.

## 7. Primary Hepatocytes

Primary cultured human hepatocytes are considered a practical and relevant liver model, enabling translatable experimentation on drug transport, metabolism, and, conversely, HDI [28]. The primary hepatocytes are extrapolated from a single donor. The limitations of this model include reduced enzyme activities resulting from the dependent decline in the expression of mRNA for major CYPs (Table 1) [35], as well as unstable cell viability, which should be determined by trypan blue exclusion or lactate dehydrogenase assay during metabolic experiments. In addition, key transporters are rapidly downregulated following hepatocyte isolation, and support matrixes may contribute artifacts [36]. Hepatocytes must be frozen and suspended in an isotonic Percoll solution before culture. After centrifugation, hepatocytes must be resuspended in a modified Krebs–Henseleit buffer (KHB), then viability testing must be performed. During the initial incubation phase, hepatocyte viability decreases significantly (personal communications). To clarify the effects of hepatocytes on metabolism, control incubations should contain the test drug in the culture medium without hepatocytes and hepatocytes alone in the absence of the test drug. As a result, the model is unsuitable for long-term herb–drug experiments and must be used in conjunction with other liver models to mimic the common practice of long-term co-administration of herbal supplements and conventional drugs.

### 7.1. HepG2/C3A Cells

C3A cells are human hepatocytes derived from hepatoblastoma-based HepG2 cells [37]. The HepG2/C3A sub-clone has improved the differentiated hepatocyte phenotype and metabolizing enzymes. Conventional cell systems rely on growing HepG2/C3A cells in 2D monolayers, often on collagen-coated rat tail dishes. It has become increasingly apparent, however, that such growth conditions do not give physiological biochemical and biomechanical cues, and as a result, human liver cells lose their phenotypes and liver-specific capabilities swiftly [38]. The molecular mechanisms governing hepatocyte dedifferentiation have been the subject of significant research efforts during the past few years. High-resolution longitudinal transcriptome investigation of primary human hepatocytes (PHH) during dedifferentiation revealed that the first phenotypic changes were evident 30 min after plating in 2D culture, with over 5000 genes being differently expressed after 4 h. Particularly affected by dedifferentiation are genes involved in complement system which include fatty acid turnover, and xenobiotic metabolism. After 24 h of monolayer culture, the expression of essential phase I and phase II enzymes and the activity of CYP1A2, CYP2C8, CYP2C9, CYP2D6, and CYP3A4, were downregulated by 90% to 99% [39]. Consequently, this indicated that, despite the model’s great potential, considerable alteration of the way cells are cultivated is necessary to improve the existing model. This led to the subsequent discovery of 3D cultured liver spheroids.

### 7.2. 3D Cultured Liver Spheroids

In recent years, a variety of 3D model systems have been developed for the study of liver function. In general, 3D cultures retain or enhance the primary hepatic functions (including the expression of drug-metabolizing enzymes) of various liver cells over several weeks of culture, so enabling long-term and repeated-dose toxicity investigations. Therefore, 3D cultured liver spheroids are perhaps a new gold standard of liver models, which have shown enhanced drug-metabolizing activities when the cells are cultured under three-dimensional conditions to form spheroids [29]. HepG2/C3A liver spheroids were first established to assess the end stage of hepatic failure. The system aimed to improve a recipient’s condition to support effective transplantation and assist in the critical step of the postoperative period, thus improving the survival rate [40]. The HepG2/C3A liver spheroids have been shown to have an enhanced life span compared to their 2D monocultures, allowing them to be cultured for long periods and demonstrating increased expression of CYP3A4 [41]. Cellular polarization, zonation, and superior liver-specific functionality have been observed in C3A liver spheroids, confirming their suitability as an *in vivo*-like liver model to study herb–drug interactions [41]. Recent studies have investigated the growth of hepatic cells under dynamic conditions within a 3D hydroscaffold integrated into a microfluidic device. Overall, these findings underline the relevance of the liver organ-on-chip model paired with a hydroscaffold in enhancing cell functions and its potential for designing a meaningful liver model for drug screening and disease research. A study by Hlengwa et al. (2019) successfully demonstrated the use of 3D cultured HepG2/C3A spheroids to assess herb–drug interactions. The study showed that the co-administration of *Lessertia frutescens* and *Echinacea purpurea* extracts affected the metabolism of ethinylestradiol in 3D cultured HepG2/C3A spheroids, validating the utility of this three-dimensional cell culture model as a predictor of hepatic herb–drug interactions in humans. Additionally, the HepG2/C3A liver spheroids were distinguished by their long-term stability and expression of all major CYPs and other key xenobiotic genes, implying that they do not need to be utilized in conjunction with other models, as is the standard for other liver models.

## 8. Liver Organ-on-Chip Models 

Liver organ-on-chip models are a type of microfluidic device that mimics the physiological and biochemical characteristics of the liver *in vitro* [10]. They consist of microfabricated channels lined with liver cells and surrounded by a matrix that mimics the extracellular environment. These models offer several advantages over traditional cell culture models and animal testing for toxicity studies and risk assessment [32]. Liver organ-on-chip models can be used to assess the toxicity of drugs, chemicals, and other substances. They allow for the measurement of specific endpoints such as drug metabolism, oxidative stress, inflammation, and cell death, which are relevant to liver toxicity [32]. These models can also be used to study the mechanisms underlying toxicity and to identify biomarkers of toxicity. One of the major advantages of liver organ-on-chip models is their ability to provide more physiologically relevant results than traditional *in vitro* models. The models can be designed to mimic the microenvironment of the liver, including the flow of blood, nutrients, and metabolites. This allows for more accurate predictions of the effects of drugs and chemicals on the liver [30]. In addition to toxicity studies, liver organ-on-chip models can also be used to study herb–drug interactions. Liver organ-on-chip models can be used to simulate these interactions and to study the effects of herbal supplements on drug metabolism and toxicity. They offer a more accurate, cost-effective, and humane alternative to traditional animal testing and cell culture models. As the technology continues to improve, we can expect to see increased adoption of liver organ-on-chip models in the pharmaceutical and biotech industries, as well as in regulatory agencies responsible for ensuring the safety of drugs and chemicals.

## 9. Discussion

The occurrence of herb–drug interaction in humans is still unpredictable, resulting in high attrition rates of herbal drug candidates in the pharmaceutical industry at the non-clinical, clinical, and post-marketing authorization stages. This is due, in part, to animal models failing to predict numerous human adverse drug reactions (ADRs), resulting in unreported herb–drug interactions during the non-clinical period of drug development. Various ways to improve the physiological relevance of hepatic *in vitro* systems are being studied to improve the prediction of human herb–drug interactions. The inclusion of additional cell types, incorporating fluid flow, and forming oxygen and nutritional gradients are all made possible by three-dimensional (3D) or microfluidic technologies, which increase differentiated cell phenotype and functionality. Innovative cell culture and tissue engineering techniques, as well as integrated endpoints, have been adopted for enhancing liver cell metabolic performance *in vitro* and are anticipated to generate more reliable data on the potential risks of pharmaceuticals [9,10,21,23,26,27,28]. Existing methods include 3D structures, flow-based cultures, co-cultures, and stem cell differentiation. A good example of such a model includes 3D bioprinting techniques. There are now accessible instances of 3D bioprinting techniques with improved *in vitro* liver cell functionality. Organoids of HepaRG and human stellate cells imitating hepatic lobules exhibited more ALB and CYP3A4 expression than monolayer cultures of HepaRG [42]. Another example includes microfluidic platforms. The combination of microfabrication techniques, such as photolithography, which is commonly used to produce computer chips, and the rapid development of tissue engineering led to the establishment and expansion of systems with dimensions in the micrometer scale for cell culture purposes, i.e., the MP or organ-on-chip (OoC) systems [10]. To overcome the shortcomings of the current liver model, powerful and innovative approaches are required as pharmacology and toxicology research gradually transitions from animal testing to more predictive *in vitro* models. When isolated, liver models retain *in vivo* liver-specific functions and their original structure to variable degrees; this, in turn, raises questions about their durability and quality decline [25]. Thus, we suggest a roadmap for herbal medicine evaluation based on fully defined, fit-for-purpose *in vitro* models, using the best of each model to ultimately contribute to more informed decision-making in the drug development and risk assessment areas. 

## 10. Conclusions

The emergence of more efficient *in vitro* liver models is a significant advancement, but there are still challenges that need to be addressed. For example, the impact of factors such as age, gender, genetics, and disease state and their effect on herb–drug interactions need to be considered, and *in vitro* hepatic models need to be designed to account for these factors. Additionally, there is a need to translate the results obtained from *in vitro* hepatic models into clinical practice and understand their implications for patient care and drug development. Despite these challenges, the future of *in vitro* hepatic models for assessing herb–drug interactions is promising. Advances in computational modelling can be used to simulate and predict herb–drug interactions and guide the development and optimization of *in vitro* hepatic models. Overall, continued development and optimization of *in vitro* hepatic models will be crucial for improving drug safety and efficacy, and the potential for more accurate and physiologically relevant models is vast.

## Figures and Tables

**Figure 1 pharmaceuticals-16-00409-f001:**
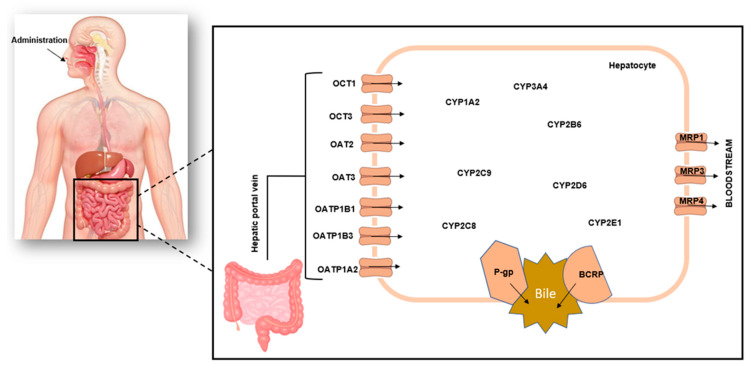
Schematic representation of hepatic drug metabolism. Orally administered drugs pass through the gastrointestinal tract to the intestines where absorption occurs and go to the liver via the hepatic portal vein, and this movement is facilitated by various drug transporters. Drug metabolism primarily takes place in the liver, but it can also happen in other organs such as the lungs, intestines, kidneys, blood, skin, and adrenals. In the liver, drugs are metabolized by cytochrome P450 enzymes. Once they have been metabolized, they enter the enterohepatic recirculation via the bile duct.

**Figure 2 pharmaceuticals-16-00409-f002:**
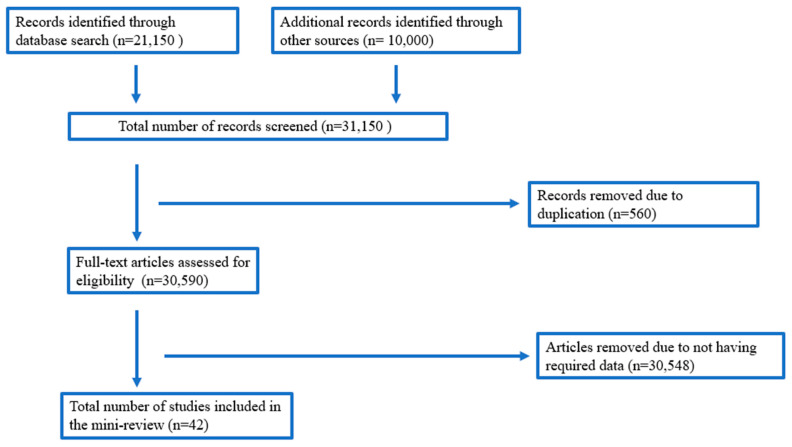
Diagrammatic representation of the literature search process.

## Data Availability

Not applicable.

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
