# Peer review of "In Vitro* Hepatic Models to Assess Herb–Drug Interactions: Approaches and Challenges"

_pharmaceuticals, 2023, doi:10.3390/ph16030409_

Round 1

Reviewer 1 Report

The review “In vitro hepatic models to assess herb-drug interactions: Approaches and challenges” was well organized and summarized. It can be considered to publish in “Pharmaceuticals” after following revisions: 

Major revisions:

1. Although the authors have their own selection criteria for the publications, I still hold that 37 references are really not enough for a review. For example, the authors claimed that they summarized the publications from 1985 to December 2022, but according to the 37 references listed by the authors, they only cited one publication from 1985, and the next one is from 1995, so the 10 years between 1985 and 1995 are blank? What is more, the authors also only cited one publication in the whole 2022, which is not enough to explain the recent research and development of the in vitro hepatic models.

2. In the conclusion section, the authors should add more contents of their own understanding and ideas, and raise more constructive suggestions for the future development prospect and direction of the in vitro hepatic model.

3. Authors should check their references more carefully. Some publications appear in the main text but were not listed in the References section at the end, such as “Gonzalez and Korzekwa, 1995” in line 124-125, “Knowles et al., 1980” and “Vouros, 1975” in line 243. This is a serious mistake. 

Minor revision:

1. The corresponding authors were not marked, and there is an extra colon between the two Emails in line 9.

2. The headings of the text were oddly numbered. There is "1. Introduction", "6. Discussion" and "7. Conclusion", but where are the 2, 3, 4, and 5?

3. Line 151: Is “phenylmethylsulfonylfluoride” a word? But it is usually written as “phenylmethylsulfonyl fluoride”. The same below.

4. Line 152: The abbreviation of “EDTA” appears for the first time here, so it should be defined.

5. Line 167: The font of “Gajula., 2022” should be consistent with the main text.

6. Line 198: The citation should be indicated after “ Dewyse et al.”

7. Line 241 and 261: Why are the both headings numbered a?

8. Figure 1: Part of words in the picture is blocked, and some words, such as "OCT1", "OCT3", etc., are not clear enough. In addition, line 298 has an extra dot.

9. Line 390: Please provide the journal name, volume number, issue number and page number of the reference.

Author Response

Dear Reviewer, 

We appreciate the time and effort that you have dedicated to providing your valuable feedback on our manuscript.  Your valuable and insightful comments have guided the improvements in the revised manuscript. We have provided a point-by-point response to your comments and concerns

Sincerely 

Dr Charity Masilela

(for authors)

Major revisions:

  1. Although the authors have their own selection criteria for the publications, I still hold that 37 references are really not enough for a review. For example, the authors claimed that they summarized the publications from 1985 to December 2022, but according to the 37 references listed by the authors, they only cited one publication from 1985, and the next one is from 1995, so the 10 years between 1985 and 1995 are blank? What is more, the authors also only cited one publication in the whole 2022, which is not enough to explain the recent research and development of the in vitro hepatic models.

Authors’ comments: Thank you for the comments. We have added additional references between the year 1985- to 1995.

  1. In the conclusion section, the authors should add more contents of their own understanding and ideas, and raise more constructive suggestions for the future development prospect and direction of the in vitro hepatic model.

Authors’ response: Thank you for these insightful comments. A sentence communicating our suggestions and future prospects has been added in the conclusion.

  1. Authors should check their references more carefully. Some publications appear in the main text but were not listed in the References section at the end, such as “Gonzalez and Korzekwa, 1995” in line 124-125, “Knowles et al., 1980” and “Vouros, 1975” in line 243. This is a serious mistake.

Authors’ response: Thank you for pointing this out. The intext references as well as the reference list have been updated.

Minor revision:

  1. The corresponding authors were not marked, and there is an extra colon between the two Emails in line 9.

Authors’ response: Thank you. The corresponding authors have all been marked  and the extra colon has been removed.

  1. The headings of the text were oddly numbered. There is "1. Introduction", "6. Discussion" and "7. Conclusion", but where are the 2, 3, 4, and 5?

Authors comment: Thank you for the comment. The numbering of headings has been amended throughout the whole document.

  1. Line 151: Is “phenylmethylsulfonylfluoride” a word? But it is usually written as “phenylmethylsulfonyl fluoride”. The same below.

Authors comment: Phenylmethylsulfonylfluoride has been amended to phenylmethylsulfonyl fluoride

  1. Line 152: The abbreviation of “EDTA” appears for the first time here, so it should be defined.

Authors comment: EDTA has been amended to Ethylenediaminetetraacetic acid

  1. Line 167: The font of “Gajula., 2022” should be consistent with the main text.

Authors comment: Thank comments. We have amended the font style of the citation, and also checked for consistency throughout the document.

  1. Line 198: The citation should be indicated after “ Dewyse et al.”

Authors comment: Thank you for the comment; however, the requested change is not clear. The sentence in line 198 ends in line 199, and we initially had two citations “Dewyse et al., 2021; Polidoro et al., 2021”.

  1. Line 241 and 261: Why are the both headings numbered a?

Authors comment: Line 241 and 261have been amended and now are numbered a. HepG2/C3A cells and b. 3D cultured liver spheroids.

  1. Figure 1: Part of words in the picture is blocked, and some words, such as "OCT1", "OCT3", etc., are not clear enough. In addition, line 298 has an extra dot.

Authors comment: Thank you for pointing this out. We have replaced our initial picture with a clearer version.

  1. Line 390: Please provide the journal name, volume number, issue number and page number of the reference.

Author comments:  The requested changes have been made.

Reviewer 2 Report

1. The percentage of people who die each year as a result of HDI should be mentioned in the introduction section. Similarly, information about liver-on-a-chip technology should be included in this section. This technology is replacing animal models and is set to become the gold standard for drug testing in the future. It should undoubtedly be included in this manuscript.

2. The authors must include a flowchart that represents the method used for the manuscript. It is easier to understand visually than it is to read it.

3. Tabla 1 should be placed closer to where it is mentioned for the first time.

4. It is essential that a section explaining liver-on-a-chip technology be included.

5. Figure #1 should have higher resolution; it is very blurry. This figure should also have numbers that represent the order of events to aid comprehension.

6. Incorporate liver-on-a-chip into Table 1.

7. Include the topic of liver-on-a-chip in the discussion and conclusion.

8. Line 261 should be labeled "b" rather than "a".

Author Response

Dear Reviewer,

We appreciate the time and effort that you  have dedicated to providing your valuable feedback on our manuscript.  indeed, your comments have improved our manuscript. We have provided a point-by-point response to your comments and concerns. We have also incorporated changes that reflect the suggestions made. 

Sincerely

Dr Charity Masilela

(For Authors)

Reviewer 2

1.The percentage of people who die each year as a result of HDI should be mentioned in the introduction section. Similarly, information about liver-on-a-chip technology should be included in this section. This technology is replacing animal models and is set to become the gold standard for drug testing in the future. It should undoubtedly be included in this manuscript.

Authors comment: Thank you for this insightful comment. However, the reviewer should note that the number of people who die from herb-drug interactions each year is difficult to determine as such cases may go unrecognized or unreported. Additionally, there is often limited information on the use of herbs and supplements, which can make it difficult to identify a specific cause of death. Nonetheless, a few sentences discussing the estimated number of deaths due to herb-drug interactions have been included in the introduction and are highlighted in red. Also, a section briefly highlighting liver-on-a-chip has been added. More details about this model are highlighted in section 8.

  1. The authors must include a flowchart that represents the method used for the manuscript. It is easier to understand visually than it is to read it.

Authors comment: thank you, this comment has been addressed (See page 4).

  1. Table 1 should be placed closer to where it is mentioned for the first time.

Authors comment: Table 1 has been moved to page 8

  1. It is essential that a section explaining liver-on-a-chip technology be included.

Authors comment: Thank you for this recommendation. A Sub-section discussing liver on chip has been added.

  1. Figure #1 should have higher resolution; it is very blurry. This figure should also have numbers that represent the order of events to aid comprehension.

Authors comments: Figure 1(now figure 2) has been replaced with a clear picture.

  1. Incorporate liver-on-a-chip into Table 1.

Authors comment: This comment has been addressed. New additions in table 1 are highlighted in red.

  1. Include the topic of liver-on-a-chip in the discussion and conclusion.

Authors comments: Thank you for this insightful comment. The discussion and conclusion have been adapted to reflect the recommendations.

  1. Line 261 should be labeled "b" rather than "a".

Authors comment: this comment has been addressed.

Reviewer 3 Report

This review is clear, comprehensive, and of general significance to the readers of this journal.

Focusing on in vitro hepatic models this mini-review discusses the existing bibliography aiming at herb-drug interactions and liver toxicity. The limitations and advantages/applications of those models are addressed throughout the manuscript. To be noticed that the study and update of these models are particularly significant, once traditional herbal medicine is used as primary health care by approximately 80% of the world population.

The introduction part covers both old and new references in a succinct way and has perfect integration of the main aspects of the theme.

This article is well written, with a good organization of the contents and a clear and pertinent methodology, particularly the search design and strategy.

The figures (only one) and tables (only one) are appropriate to the discussion of the theme, we believe that the quality of the manuscript will be increased with some creative author’s effort in this aspect. We kindly suggest the authors improve the manuscript in this aspect, to match the excellence of Pharmaceuticals Journal.

The cited core references are recent and appropriate to the discussion.

Although the application of in vitro models in the detection of therapeutic effects is mentioned as an object of discussion, in the abstract (L17) and last paragraph of the Introduction (L79), this discussion seems nowhere to be found. Please, dear authors, clarify why “therapeutic effects” was mentioned in L17 and L79.

Author Response

Dear Reviewer, 

Dear Editor,

We appreciate the time and effort that you have dedicated to providing your valuable feedback on our manuscript.  Your valuable and insightful comments have guided the improvements in the revised manuscript. We have incorporated changes that reflect the suggestions provided. 

Sincerely

Dr Charity Masilela

(For Authors)

Reviewer 3

This review is clear, comprehensive, and of general significance to the readers of this journal.

Focusing on in vitro hepatic models this mini-review discusses the existing bibliography aiming at herb-drug interactions and liver toxicity. The limitations and advantages/applications of those models are addressed throughout the manuscript. To be noticed that the study and update of these models are particularly significant, once traditional herbal medicine is used as primary health care by approximately 80% of the world population.

The introduction part covers both old and new references in a succinct way and has perfect integration of the main aspects of the theme.

This article is well written, with a good organization of the contents and a clear and pertinent methodology, particularly the search design and strategy.

The figures (only one) and tables (only one) are appropriate to the discussion of the theme, we believe that the quality of the manuscript will be increased with some creative author’s effort in this aspect. We kindly suggest the authors improve the manuscript in this aspect, to match the excellence of Pharmaceuticals Journal.

Authors comment: Thank you for this recommendation. The location of the table has been changed to be closer to the text where it is first mentioned. Also, we have improved the quality of the diagram.

The cited core references are recent and appropriate to the discussion.

Although the application of in vitro models in the detection of therapeutic effects is mentioned as an object of discussion, in the abstract (L17) and last paragraph of the Introduction (L79), this discussion seems nowhere to be found. Please, dear authors, clarify why “therapeutic effects” was mentioned in L17 and L79.

Authors comments: Thank you for the comments. The authors recognise that the term “therapeutic effects” was used out of context. The phrase has been removed in the above-mentioned locations.

Round 2

Reviewer 1 Report

The manuscript was well revised accordingly.